# Coherent Multi-Dwell Processing of Un-Synchronized Dwells for High Velocity Estimation and Super-Resolution in Radar

**Benzion Levy [1,*], Lior Maman [2], Shlomi Shvartzman [2] and Yosef Pinhasi [1]**

[1] Faculty of Engineering, Ariel University, Ariel 40700, Israel
[2] Faculty of Engineering, Tel Aviv University, Tel Aviv 39040, Israel
* Correspondence: benzion.levy@msmail.ariel.ac.il; Tel.: +972-3-958-5844

**Abstract:** This paper describes a coherent multi-dwell processing (CMDP) method for high velocity estimation and super-resolution in search and track, while search (TWS) radar modes use an unconventional signal processing algorithm that exploits multi-dwell transmissions. The existence of the multi-dwell waveform is necessary for visibility needs by un-folding the target's velocity and range ambiguity and is proposed to be utilized for high velocity estimation and super-resolution. In this paper, the proposed scheme is shown to result in improved velocity estimation and doppler resolution performance for un-ambiguous targets in comparison to classical radar processing. The processing concept uses the same transmitted waveform (WF) and time duration without the need to increase the time on target (TOT) through sophisticated coherent concatenation of the received dwells with velocity compensation between the dwells. The phase compensation in receive mode is implemented for each target according to its characteristics, which means that target velocities are estimated in each dwell separately. The notable result of the CMDP is the linear doppler resolution improvement obtained with the given search resources and without knowing the target characteristics in advance or the dwell delay time. Other possible benefits of this process are the ability to achieve larger detection ranges and high-angle measurement precisions in search mode due to the higher signal-to-noise ratio (SNR) of the extended dwell and the ability to track more targets due to efficient time and resource management. An outstanding opportunity to exploit the CMDP is by combining missions in phased array (PA) radars, meeting the multi-objective needs of both high spatial scan rates for illuminating the target and high doppler estimation and resolution performance.

**Keywords:** phased array; AESA; super-resolution; radar; DBF; doppler; visibility map; TWS; revisit time; PRF; TOT; estimation

## 1. Introduction

Phased-array (PA) technology is conventionally used in the military for radar and satellite applications [1]. This technology has many performance advantages in comparison to conventional mechanical scan antennas [2], such as multi-functionality and flexibility in using special and dedicated beams with adequate waveform parameters according to the target demand. The performance of radar is widely affected by many technical parameters [3], including the spatial scanning pattern and waveform parameters.

It is common to implement PA radars with multi-functionality by using resource management allocations in search and tracking missions. The search beams are scheduled according to the revisit time constraints, meeting the multi-objective needs of both illuminating the target and obtaining the required doppler estimation and resolution that are directly derived from the SNR and the dwell duration [4]. This tradeoff between the spatial coverage, subjected to the revisit time constraints, and the desired velocity resolution, which is upper bounded by the FFT windowing of the dwell duration, is a key consideration in radar design [5]. Usually, in AESA radars, the design principle of the search mode is calculated according to the nearest and farthest coverage, where the nearest target

defines the revisit time constraints and the farthest target defines the minimal SNR. Such a typical radar search mode is necessarily designed with a multi-PRI waveform for high visibility performance by solving the range and doppler ambiguities caused by radar parameters. However, the separate processing of each dwell does not utilize the whole burst transmission time to obtain the best velocity resolution and estimation.

Super resolution velocity and high velocity estimation have been popular in the academic world since they provide higher resolution than classical burst processing based on dwell's interpolation techniques [6,7]. Considerable research has been conducted on velocity super-resolution using multiple signal classification (MUSIC) [8,9] algorithms and orthogonal frequency-deviation multiplexing (OFDM) [10,11], taking into account the known number of targets. Moreover, the possibilities of reducing the average time on target (TOT) in pulse-doppler radar by sub-Nyquist techniques, including compressed sensing (CS) optimization [12], have also been carried out over the years, finding theoretical bounds. Some other studies used random PRIs to alleviate range and doppler ambiguities as well as to enhance electronic counter-countermeasure (ECCM) capabilities [13].

The present paper proposes a new approach for coherent processing across multiple dwells in radar for single and multiple targets, assuming a given scanning pattern, transmitted waveform and burst duration, using an analytical solution that is not affected by the dwell's incoherency. Consequently, improved doppler resolution performance and SNR are achieved without the need to increase the TOT.

This paper is divided into three sections. In the first section, we state the theory and background of velocity super-resolution and estimation, the problem formulation and describe the upper bound performances. In the second section, we show a simulation for improving the velocity estimation of a single target by achieving a higher SNR. In the third section, an algorithm and simulation of velocity super-resolution will be presented, including numerous targets with varied velocities and incoherent dwells. Finally, the performances will be compared, and suggested radar implementations will be discussed, as well as further research.

## 2. Theoretical Background

### 2.1. Multi-PRI in Classical Radar Processing

The existence of the multi-dwell waveform in search mode is necessary for visibility needs, obtaining high range and velocity coverage by un-folding the target's velocity and range ambiguities [14,15]. In addition, the radar has blind speeds and blind ranges that depend on the transmitted waveform. Figure 1 shows the visibility maps of two PRIs with blind markings that occur at multiples of the PRI and 1/PRI for range and velocity, respectively, due to the MTI filter, designated for clutter rejection, and the inability to receive during pulse transmission.

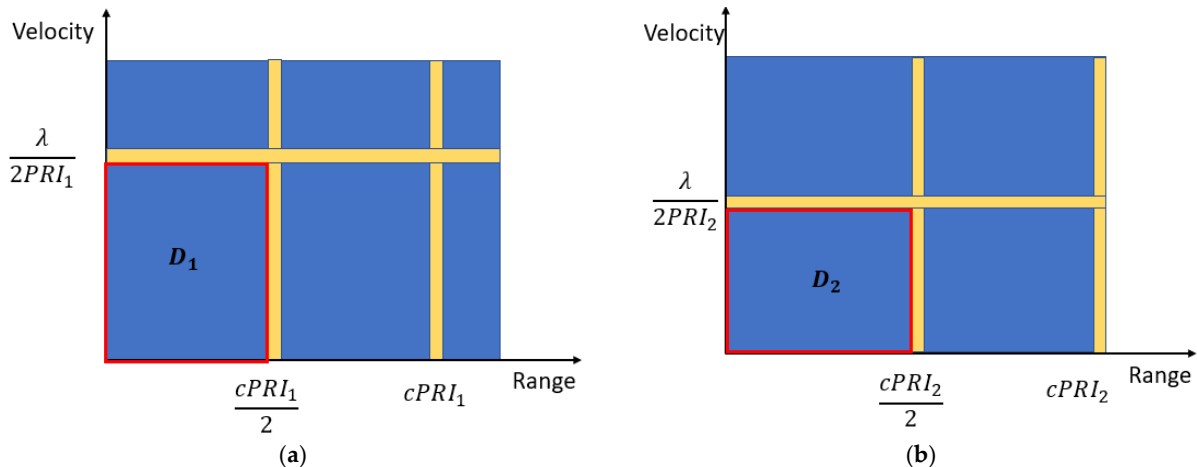

**Figure 1.** Visibility map of (**a**) $PRI_1$; (**b**) $PRI_2$.

The common un-ambiguous region is the common domain of the un-ambiguous ranges and dopplers of all the dwells of the waveform.

$$D_{unambiguous} = \bigcap_{i=1}^{M} D_i$$

where $D_i$ is the un-ambiguous visibility domain of $PRI_i$ and $M$ is the number of dwells.

Plot detection probability is a function of the dwell's SNR and the false alarm, where the SNR of a plot is dependent on the coherent dwell duration $T_{dwell}$ according to the following radar equation [16].

$$SNR = \frac{P_t \cdot DC \cdot G_T G_R \lambda^2 \sigma \cdot T_{dwell}}{(4\pi)^3 R^4 \cdot k_b T_E \cdot NF} \tag{1}$$

where $P_t$ is the peak transmitted power, $DC$ is the duty cycle, $G_t$, $and$ $G_r$ are the antenna's transmitted and received gain, respectively, $\sigma$ is the radar cross-section (RCS), $T_{dwell}$ is the coherent integration duration, $R$ is the range, $K_b$ is the Boltzmann constant, $T_E$ is the absolute temperature and $NF$ is the noise figure.

Coherent integration is typically used in radars, which means that both the target's radar cross-section (RCS) and velocity are constant during the dwell as well as the range from the target.

The SNR of the target in the dwell is affected by both the match filter (MF), which is the correlation of the signal in the range axis, and the FFT, which is the processing in the velocity axis.

Systematically, multiple dwells are usually non-coherently integrated for cumulative target detection [17,18].

### 2.2. Theoretical Formulation

We consider the radar in Figure 2, with a waveform consisting of $M$ consecutive dwells, as illustrated in Figure 3.

The $m$th dwell contains comb of $N_m$ RF pulses of width $\tau_m$, appearing at time intervals $T_m$ (which is the PRI within the $m$th dwell):

$$\widetilde{E_m}(t) = A_T \, e^{j2\pi f_c t} \text{rect}\left(\frac{t}{\tau_m}\right) * \sum_{n=0}^{N_m-1} \delta(t - nT_m) \tag{2}$$

Here, $A_T$ is the amplitude, $f_c$ is the carrier frequency of the wave and

$$\text{rect}\left(\frac{t}{\tau_m}\right) \equiv \left\{ \begin{array}{ll} 1, & 0 < t \leq \tau_m \\ 0, & \tau_m < t < T_m \end{array} \right.$$

is a rectangular pulse with a temporal duration of $\tau_m$, $\delta(t)$ denotes the Dirac's delta function and $*$ is a convolution. Given a radar burst composed of $M$ dwells with different PRIs, as depicted in Figure 3, the transmitted waveform is then:

$$\widetilde{E}_T(t) = \sum_{m=1}^{M} \widetilde{E}_m(t) * \delta[t - (m-1) \cdot N_{m-1} T_{m-1}] =$$

$$= A_T \, e^{j2\pi f_c t} \underbrace{\sum_{m=1}^{M} \left[ \text{rect}\left(\frac{t}{\tau_m}\right) * \sum_{n=0}^{N_m-1} \delta(t - nT_m) \right] * \delta[t - (m-1) \cdot N_{m-1} T_{m-1}]}_{W(t)} \tag{3}$$

We define the transmitted pulse train as:

$$W(t) =$$

$$= \text{rect}\left(\frac{t}{\tau_1}\right) * \sum_{n=0}^{N_1-1} \delta(t - nT_1)+$$

$$+\text{rect}\left(\frac{t}{\tau_2}\right) * \sum_{n=0}^{N_2-1} \delta[t - (N_1 T_1 + nT_2)]+$$

$$\cdot$$
$$\cdot$$
$$\cdot$$

$$+\text{rect}\left(\frac{t}{\tau_M}\right) * \sum_{n=0}^{N_M-1} \delta\left[t - \left(\sum_{i=1}^{M-1} N_i T_i + nT_M\right)\right] \tag{4}$$

Note that the transmission holds for the time interval $t \in [0, N_1 T_1 + N_2 T_2 + \ldots + N_M T_M]$. The received signal is the reflected version of the transmitted waveform scattered from a target located at a range $R(t)$:

$$\widetilde{E}_R(t) = A_R e^{j2\pi f_c[t - \frac{2R(t)}{c}]} \cdot W\left[t - \frac{2R(t)}{c}\right] \tag{5}$$

where $c \cong 2.998 \cdot 10^8 \frac{m}{s}$ is the speed of light.

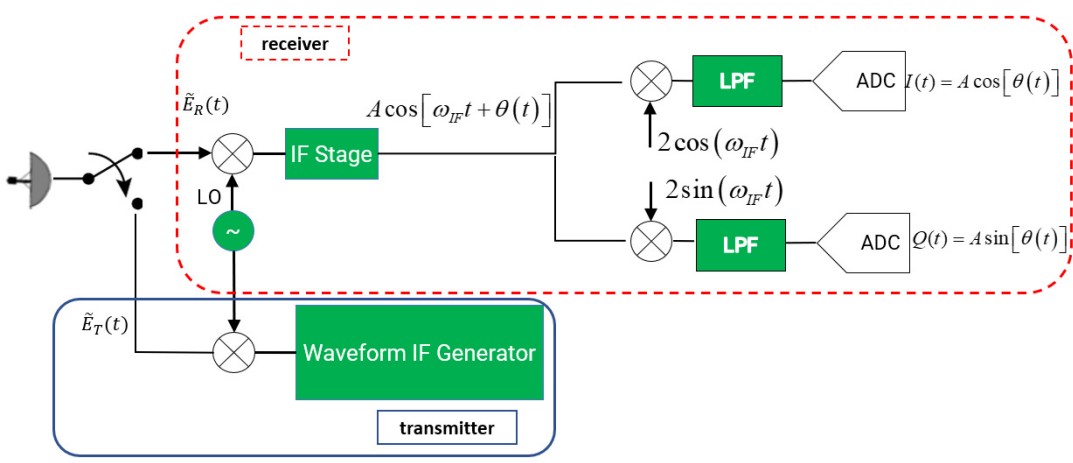

**Figure 2.** Radar transceiver.

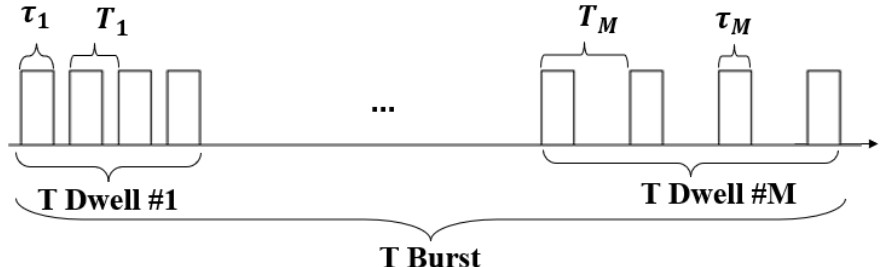

**Figure 3.** Radar burst waveform.

The waveform characteristics are shown in Table 1.

**Table 1.** Waveform characteristics.

| Denotation | Symbol | Units |
|---|---|---|
| Carrier frequency | $f_c = 3$ | GHz |
| Speed of light | $c$ | m/s |
| Wavelength | $\lambda = \frac{c}{f_c}$ | m |
| Number of dwells | $M$ | - |
| Amplitude of the doppler signal from dwell m | $A_m$ | - |
| Number of pulses in dwell $m$ | $N_m$ | - |
| Pulse repetition interval of dwell $m$ | $T_m$ | s |
| Pulse duration in dwell $m$ | $\tau_m$ | s |
| Duty cycle of dwell $m$ | $\frac{\tau_m}{T_m}$ | - |
| Time on target of dwell $m$ | $T_{\mathrm{dwell}_i} = N_m \cdot T_m$ | s |
| Burst duration | $T_{\mathrm{burst}} = \sum\limits_{m=1}^{M} N_m \cdot T_m$ | s |

Figure 4 shows a classical range-pulse map, expressing the received power from each pulse in all the ranges during the coherent single dwell, where the x-axis represents the "fast-time" and the y-axis represents the "slow-time". Each received pulse is downconverted, matched and filtered. The signal peak on the x-axis expresses the target range and we can easily see that the target exists in the range of 3 km.

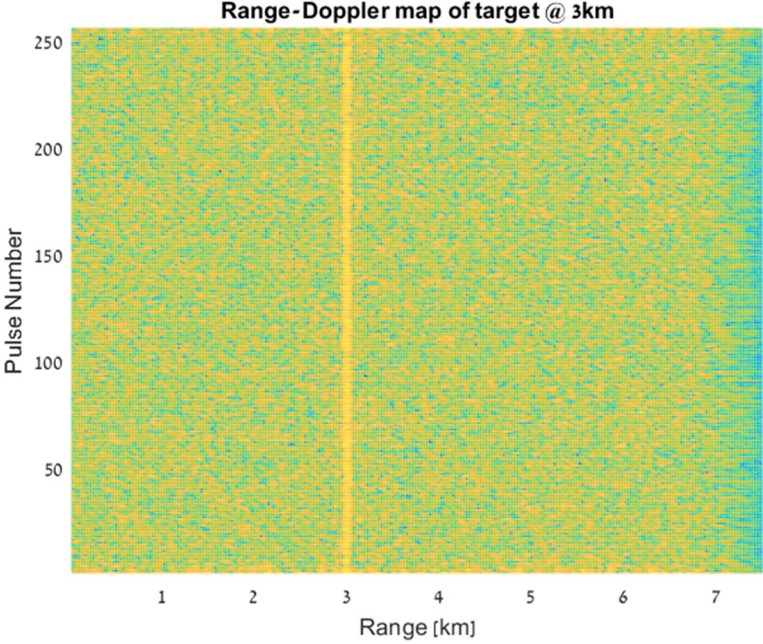

**Figure 4.** Range-pulse map after matched filter processing in the fast time.

However, the detection process has not yet ended, and the velocity also needs to be determined with sufficient accuracy. This is performed by applying FFT across the y-axis and achieving the range-doppler (RD) map, from which the detected plot will later be derived using CFAR methods.

This paper will focus on the velocity accuracy and velocity resolution that are achieved by advanced signal processing on the range-pulse map and across the multi-dwell maps.

For a given range, the doppler frequency shifts of each received dwell were also detected. The complex amplitude is given by:

$$\widetilde{V}(t) = Ae^{j2\pi f_c \cdot \frac{2R(t)}{c}} = Ae^{j\frac{2\pi}{\lambda} \cdot 2R(t)} \tag{6}$$

where $\lambda = c/f_c$ is the carrier wavelength and $A$ is a constant amplitude. The instantaneous frequency of the IF signal is given by the doppler shift.

$$f_d(t) = \frac{2}{\lambda} \cdot \frac{d}{dt} R(t) = \frac{2v_r(t)}{\lambda}$$

where $v_r(t)$ is the radial velocity. Assuming a constant target velocity $v_r$ during the coherent burst, the range within the burst can be written as:

$$R(t) = R_0 + v_r \cdot t$$

The resulting signal at the matched filter output is a sinusoidal tone at a constant frequency $f_d$ and phase $\varphi_0 = 4\pi R_0/\lambda$.

$$\widetilde{V}(t) = A\, e^{j(2\pi f_d t + \varphi_0)}$$

The detected tone is sampled at discrete times related to the individual dwell PRI, as given by the following vector:

$$\vec{t} = \left\{ \underbrace{[0,...N_1 - 1] \cdot T_1}_{\text{1st dwell}}, \cdots, \underbrace{\sum_{i=1}^{m-1} N_i T_i + [0,...,N_m - 1] \cdot T_m}_{\text{m}^{\text{th}} \text{ dwell}}, \cdots, \underbrace{\sum_{i=1}^{M-1} N_i T_i + [0,...,N_M - 1] \cdot T_M}_{\text{M}^{\text{th}} \text{ dwell}} \right\} \tag{7}$$

For instance, the resulting samples of the first dwell, where the duration of the pulses is $T_1$, are:

$$\widetilde{V}_1[n] = A_1 e^{j\varphi_0} e^{j2\pi f_d n T_1} = \widetilde{I}(nT_1) + j\widetilde{Q}(nT_1) \tag{8}$$

for $n = 0 \ldots N_1 - 1$. Here, $\widetilde{I}(nT_1)$ and $\widetilde{Q}(nT_1)$ are the in-phase and quadrature components of the $n$th sample, respectively, as illustrated in Figure 2. $A$ is the amplitude of the signal after the matched filter.

We obtained M non-uniform doppler sub-groups that cannot be processed together to derive the correct doppler frequency with a simple FFT. For this reason, in classical search radars, the dwells are processed separately, as shown in Figure 2. The concatenated vector that should be coherently processed is:

$$\underline{\widetilde{V}} = \left[ \widetilde{V}_1; \widetilde{V}_2; \cdots ; \widetilde{V}_M \right] \tag{9}$$

The significant disadvantage of separated dwell processing is the limitation of the doppler resolution, which is derived from the coherent $T_{dwell}$ of a single dwell. The velocity resolution obtained from a single $m$th dwell is bounded by:

$$\begin{aligned} \Delta f_d &= \frac{2\Delta v_r}{\lambda} = \frac{PRF}{N_m} = \frac{1}{N_m T_m} = \frac{1}{T_{dwell_m}} \\ \Delta v_r(m) &= \frac{\lambda/2}{N_m T_m} = \frac{\lambda}{2 T_{dwell_m}} \end{aligned} \tag{10}$$

The doppler resolution is achieved from the Fourier transform of signal in the 'slow time' (across the pulse axis). In practice, the FFT is performed on a bounded duration (dwell) that is equivalent to the time-window. Thus, a bounded doppler resolution is accepted with a resolution of $1/T_{dwell}$, since it is the zero point of the sinc function in the frequency domain.

In the classical approach, the resolution may be improved by increasing the dwell duration $T_{dwell}$, which means that the target is illuminated for a longer time.

In Chapter 3, a description of a coherent multi-dwell processing (CMDP) technique for a single target will be presented. The generalization of the CMDP for multiple targets is presented in Chapter 4, making it possible to process the whole waveform altogether while achieving both velocity super-resolution and high-velocity estimation.

## 3. CMDP for Single Target Estimation

CMDP algorithms exploit multi-PRI transmission by artificially producing an equal PRI. In our approach, a CMDP algorithm for phase compensation will be introduced, assuming a known velocity and coherency between the dwells. The known velocity can be evaluated from any single dwell, and the idea of using it is popular in various radar techniques, such as MIMO radars [19] and stepped frequency modulated (SFM) radars [20].

### 3.1. CMDP Model

We demonstrate the technique, considering $M = 3$ dwells, with three different PRIs, one for each dwell, as follows:

$$
\begin{aligned}
\widetilde{\underline{V}}_1 &= A_1 e^{j\frac{4\pi}{\lambda} v_r \vec{t_1}}, \vec{t_1} = [0, T_1, 2T_1, \cdots, (N_1 - 1)T_1] \\
\widetilde{\underline{V}}_2 &= A_2 e^{j\frac{4\pi}{\lambda} v_r \vec{t_2}}, \vec{t_2} = (N_1 - 1)T_1 + [T_2, 2T_2, \cdots, N_2 T_2] \\
\widetilde{\underline{V}}_3 &= A_3 e^{j\frac{4\pi}{\lambda} v_r \vec{t_3}}, \vec{t_3} = (N_1 - 1)T_1 + N_2 T_2 + [T_3, 2T_3, \cdots, N_3 T_3]
\end{aligned}
\tag{11}
$$

The three vectors are concatenated, as in (9), to obtain a vector with $N_1 + N_2 + N_3$ elements:

$$
\widetilde{\underline{V}} = \left[ \widetilde{\underline{V}}_1; \widetilde{\underline{V}}_2; \widetilde{\underline{V}}_3 \right]
$$

For simplicity and without a lack of generalization, we assume that $T_{dwell}$ and $DC$ in different dwells are equal to preserve equal SNR and accuracy per dwell.

The real scenario model consists of a signal mixed with noise; hence:

$$
\widetilde{\underline{S}} = \widetilde{\underline{V}} + \widetilde{\underline{\varepsilon}}
$$

where $\widetilde{\underline{\varepsilon}} \sim CN\left(\underline{0}, \sigma^2 I_{\sum_{m=1}^{M} N_m}\right)$, denoting the single pulse SNR for the target's sample.

The FFT will be, with respect to sampling period $T_1$ and by setting the appropriate $\widetilde{a}_m$ coefficient, applicable despite the non-uniform PRIs. For M dwells, the expression for the FFT is:

$$
\widetilde{X}[k] = \sum_{m=1}^{M} \widetilde{a}_m \cdot \left[ \sum_{n=(m-1)\cdot N_m}^{mN_m - 1} \widetilde{\underline{S}}(n) e^{-j\frac{2\pi}{M\cdot N} n\cdot k} \right] + \widetilde{\underline{\varepsilon}}
\tag{12}
$$

For M = 3:

$$
\text{Denote } N = N_1 + N_2 + N_3
$$

$$
\widetilde{X}[k] = \sum_{n=0}^{N_1 - 1} \widetilde{a}_1 e^{j\frac{4\pi}{\lambda} v_r T_1 n} e^{-j\frac{2\pi}{N} nk} + \sum_{n=N_1}^{N_1 + N_2 - 1} \widetilde{a}_2 e^{j\frac{4\pi}{\lambda} v_r T_2 n} e^{-j\frac{2\pi}{N} nk} + \sum_{n=N_1 + N_2}^{N-1} \widetilde{a}_3 e^{j\frac{4\pi}{\lambda} v_r T_3 n} e^{-j\frac{2\pi}{N} nk} + \widetilde{\underline{\varepsilon}}
$$

where the coefficients $\widetilde{a}_m$ are set to be:

$$
\widetilde{a}_m = \begin{cases} 1, & t \in [0, (N_1 - 1)T_1] \\ e^{j\frac{4\pi}{\lambda} v_r (T_1 - T_2)n}, & t \in (N_1 - 1)T_1 + [T_2, N_2 T_2] \\ e^{j\frac{4\pi}{\lambda} v_r (T_1 - T_3)n}, & t \in (N_1 - 1)T_1 + N_2 T_2 + [T_3, N_3 T_3] \end{cases}
\tag{13}
$$

The final expression for the FFT is:

$$
\widetilde{X}[k] = \sum_{n=0}^{N-1} e^{j\frac{4\pi}{\lambda} v_r T_1 n} e^{-j\frac{2\pi}{N} nk} = \sum_{n=0}^{N-1} e^{jn\left(\frac{4\pi}{\lambda} v_r T_1 - \frac{2\pi}{N} k\right)} + \widetilde{\underline{\varepsilon}}
\tag{14}
$$

It can be shown from Equation (14) that by setting the appropriate coefficients we derived an expression with a single PRI and consequently the FFT is applicable over the multi-dwell.

We demonstrate the technique for a moving target with a radial velocity of $v_r = 5 \ [m/s]$. The radar transmits $N = 128$ pulses per $M = 3$ dwells. The respective PRIs are $T_1 = 140 \ \mu$ sec;

$T_2 = 120 \, \mu \, \sec$; $T_3 = 100 \, \mu \, \sec$; SNR = 0 dB. Figure 5 presents a comparison between the velocity estimated when a single dwell is processed and when multiple dwells ($M = 3$) are coherently processed in our CMDP algorithm.

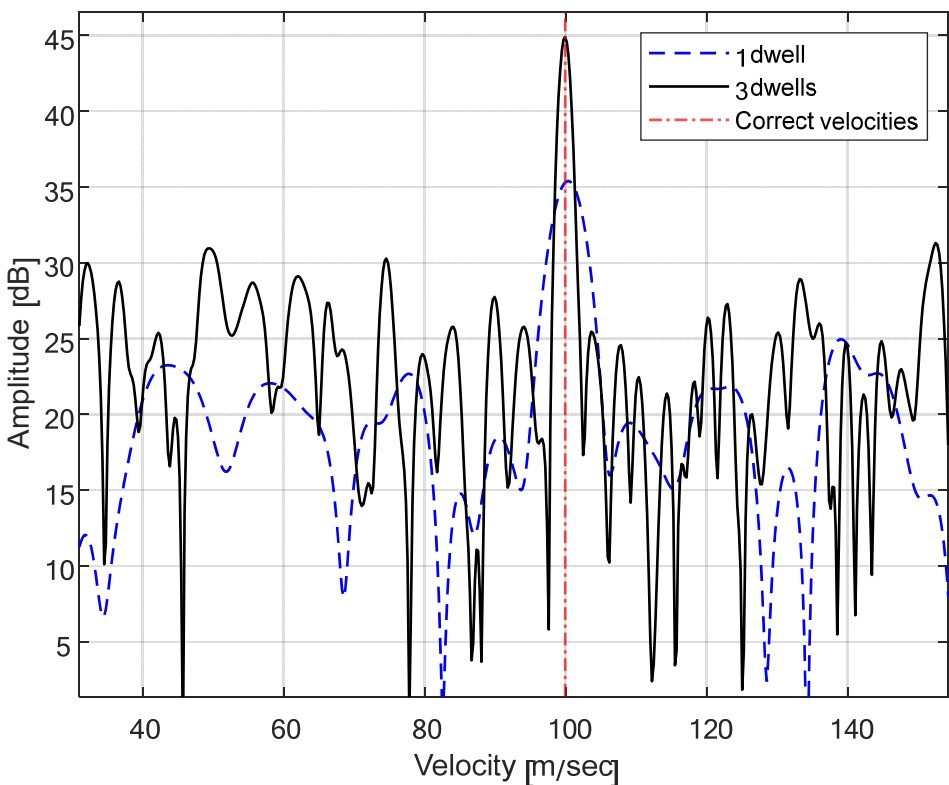

**Figure 5.** Velocity estimation for single dwell and concatenated 3 dwells.

An inspection of Figure 5 reveals an improved intensity, which is expressed as an increased SNR and consequently better radar measurement accuracies.

### 3.2. CMDP vs. Interpolation for Un-synchronized Dwells

In this section, we compare the sinc interpolation to the CMDP. The sinc interpolation method is used for coherent dwell concatenation by artificially converting the different PRIs sampled along the multi-dwell into an equivalent uniform sampling [6,7].

The interpolation method has severe drawbacks when the dwells are not synchronized and have additional delay time $\Delta T$ between them, while the CMDP is much less sensitive to time delay since the method estimates the velocity of each dwell and decreases it from the true velocity. Hence, the processing of each dwell is now performed in the baseband ($\widetilde{\underline{V}}_{1,BB}$), as described below:

$$
\begin{aligned}
\widetilde{\underline{V}}_{1,BB} &= A_1 e^{j\frac{4\pi}{\lambda}(v_r - \hat{v}_r)\vec{t_1}}, \vec{t_1} = [0, T_1, 2T_1, \cdots, (N_1 - 1)T_1] \\
\widetilde{\underline{V}}_{2,BB} &= A_2 e^{j\frac{4\pi}{\lambda}(v_r - \hat{v}_r)\vec{t_2}}, \vec{t_2} = \Delta T + (N_1 - 1)T_1 + [T_2, 2T_2, \cdots, N_2T_2]
\end{aligned}
\tag{15}
$$

Therefore, because the process is in the baseband and the frequencies are low, $\Delta T$ has less influence and after concatenating these two dwells into one longer signal, it effectively behaves similar to a single dwell.

However, in the interpolation method, two peaks may arise in the velocity spectrum. Let us assume that we have two different PRIs and delay times, then without the loss of generality, we assume $T_1 > T_2$ and denote the signals as:

$$\begin{aligned}
\underline{\widetilde{V}}_1 &= A_1 e^{j\frac{4\pi}{\lambda} v_r \vec{t_1}}, \vec{t_1} = [0, T_1, 2T_1, \cdots, (N_1-1)T_1] \\
\underline{\widetilde{V}}_2 &= A_2 e^{j\frac{4\pi}{\lambda} v_r \vec{t_2}}, \vec{t_2} = \Delta T + (N_1-1)T_1 + [T_2, 2T_2, \cdots, N_2 T_2]
\end{aligned} \tag{16}$$

We interpolate $\widetilde{V}_1$ to a new times vector, sampled from $T_2$ and reduce the interpolation error by using the sinc interpolation. The new times vector is:

$$\vec{t_1}^{\,\text{interp}} = \left[ 0, 1, \cdots, \text{round}\left( N_1 \frac{T_1}{T_2} \right) \right] T_2 \tag{17}$$

The corresponding $\widetilde{V}_1^{\text{interp}}$ is a resampled version of $\widetilde{V}_1$ at $\vec{t_1}^{\,\text{interp}}$. After concatenating $\widetilde{V}_1^{\text{interp}}$ and $\widetilde{V}_2$ we get:

$$\widetilde{V} = \left[ \widetilde{V}_1^{\text{interp}}; \widetilde{V}_2 \right] \tag{18}$$

Finally, to find the velocity we perform FFT on the concatenated signal $\widetilde{V}$ and see an extra peak, which is a result of the time delay $\Delta T$.

Figure 6 displays a comparison between the interpolation and CMDP methods for a single target with $v_r = 100\ [m/s]; \Delta T = 10T_1; \text{SNR}_{\text{pulse}} = 0$ dB.

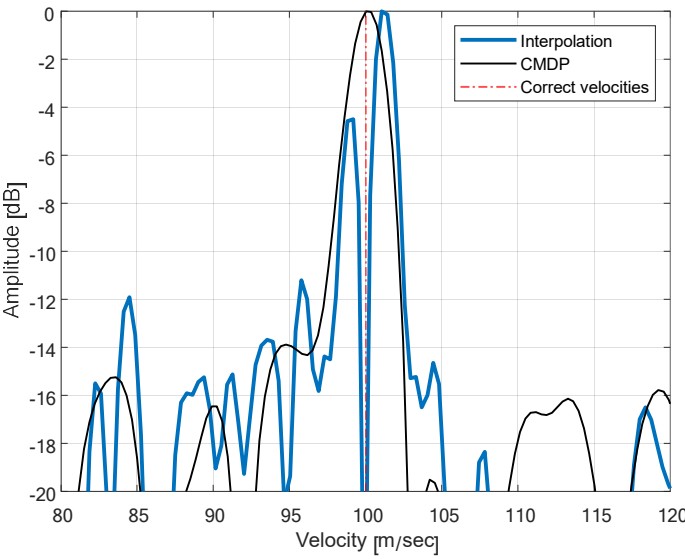

**Figure 6.** CMDP vs. interpolation for un-synchronized dwells.

In Figure 6, we see the deceptive two peaks with estimation error by the interpolation method in comparison to the analytic method of CMDP.

Similarly, phase compensation in Rx could be implemented for multiple targets, for which velocities had been detected and measured by the single dwell. Furthermore, multi-resolution could be achieved for each target according to the desired goal by setting a compatible multi-dwell duration.

## 4. CMDP Algorithm for Multiple Target Estimation and Resolution

In this section, the CMDP is developed for the general multi-target case based on a different approach to the CMDP algorithm. The algorithm will show the benefits of velocity estimation as well as doppler resolution and SNR improvement.

*CMDP Algorithms*

We propose an iterative method to detect targets from multi-PRI dwells, as displays in the flow chart in Figure 7. Starting with the first dwell, we detect all target velocities that passed a preset threshold and put all of them in a list L. Then, we run over all the targets and fix all M dwells with the same velocity (one target is fixed, other targets potentially ruined); afterwards, we concatenate all the fixed dwells and detect again, now from M dwells, and ask whether a new target is revealed. It is important to note that at this stage, the targets are revealed because of a better resolution with M dwells. If only one target exists, go to the next target. Otherwise, more than one target is revealed, hence removing the current velocity and adding (at the end of the list) all the newly detected targets and trying to improve them again.

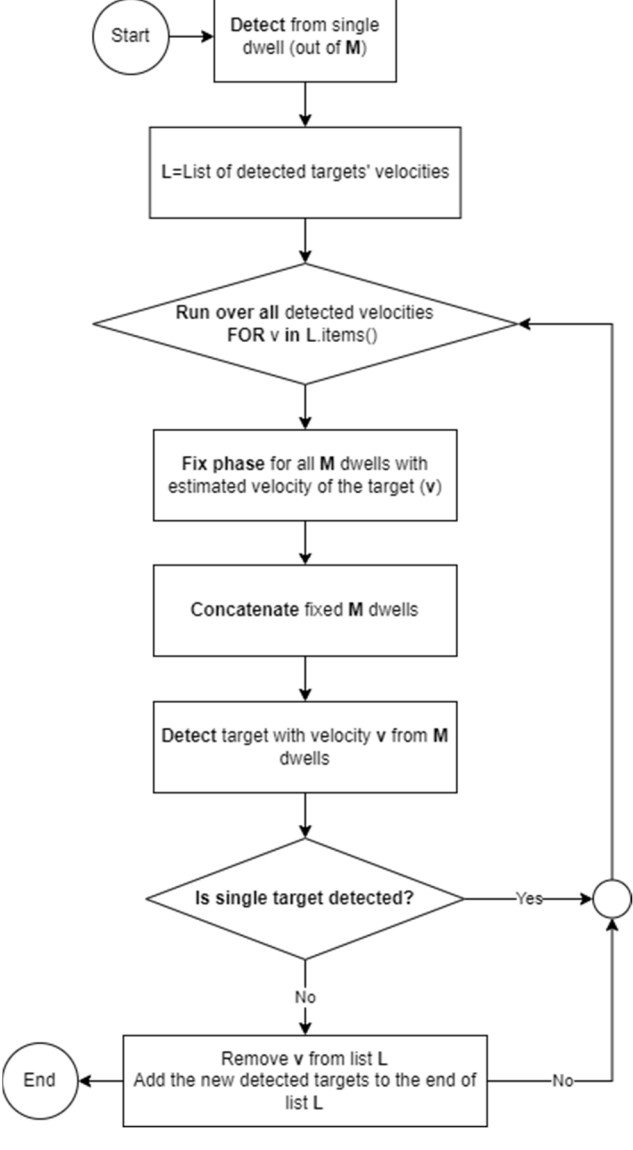

**Figure 7.** CMDP Algorithm.

To estimate the target velocity from the multiple dwells, we fuse the estimated velocities from each dwell. In the general case of different velocity accuracies, we have a fusion estimation that is given by:

$$\hat{v}_r = \frac{\sum_{m=1}^{M} \frac{\hat{v}_{rm}}{\sigma_{v_m}^2}}{\sum_{m=1}^{M} \frac{1}{\sigma_{v_{rm}}^2}} \tag{19}$$

where $\hat{v}_r$ is the weighted average estimation of the multi-dwell, $\hat{v}_{rm}$ is the estimated velocity from the $m$th dwell and the standard deviation $\sigma_{v_{r_m}}$ is [21]:

$$\sigma_{v_{r_m}} = \frac{\lambda}{2 T_{\text{dwell}_m}} \frac{1}{\sqrt{k \cdot SNR}}; k = const(usuall \sim 2:3) \tag{20}$$

We demonstrate the CMDP algorithm for multiple targets with radial velocities of $v_r = (100, 101, 120, 135) \, \text{m/sec}, T_1 = 140 \, \mu\text{sec}; T_2 = 120 \, \mu\text{sec}; T_3 = 100 \, \mu\text{sec}; SNR_{pulse} = 0 \, \text{dB}$ and use Blackman windowing in CMDP. For equal velocity error in each dwell, $T_{\text{dwell}_m}$ and DC are the same. Thus, from (19), the weighed velocity is a simple average.

Figure 8 shows a comparison between single and multi-dwell processing.

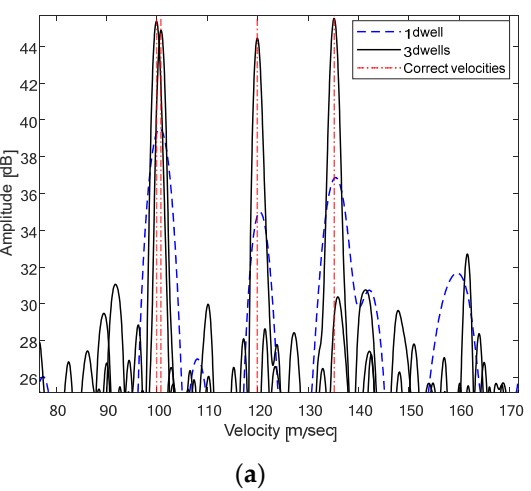

(**a**)

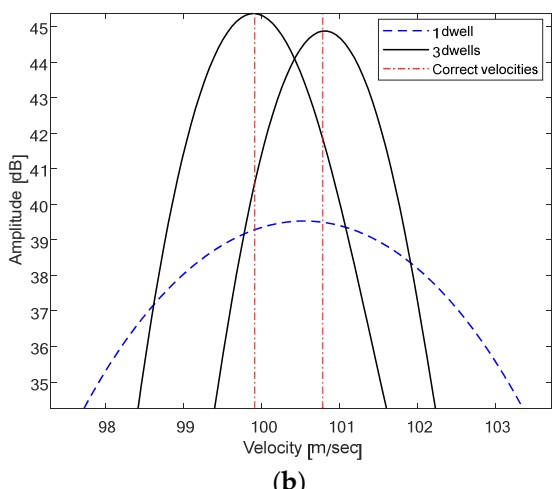

(**b**)

**Figure 8.** CMDP simulation results. (**a**) Shows achieving both higher velocity estimation and improved SNR. (**b**) Shows that an additional real target has been detected as a result of the super-resolution.

## 5. Conclusions

In this paper, we showed that velocity super-resolution and high estimation in search radars are achievable by using an un-conventional signal processing algorithm that exploits multi-dwells transmission. The notable result of the CMDP is the linear doppler resolution improvement for radars with un-synchronized dwells, obtained by smart coherent concatenation of the received dwells, with delay time compensation and without prior knowledge of the velocities. Actually, a diverse resolution could be achieved by adapting appropriate coherent time frames and producing the desired resolution.

Optional applications of using this algorithm are in TWS radars by saving the search resources and therefore increasing the number of tracked target capabilities. Moreover, computational resources are saved for high doppler resolutions with long burst processing by alleviating the need to maintain coherency between all dwells. An outstanding opportunity to exploit the CMDP is by combining a drone mission [22] in phased array (PA) radars, meeting the multi-objective needs of both a high spatial scan rate for illuminating the target and high doppler resolution performances. In addition, an SNR improvement is achieved and consequently better range detection, as well as radar parameter accuracies, such as velocity and angle.

Further research could be conducted in the case of an incoherent range using key-stone techniques [23] for target focusing and by expanding the high-resolution algorithm to ambiguous targets.

**Author Contributions:** B.L., L.M. and S.S. wrote and performed the simulations; B.L., L.M., S.S. and Y.P. conceived the theory; B.L., L.M. and S.S. wrote the paper. All authors have read and agreed to the published version of the manuscript.

**Funding:** This research received no external funding.

**Data Availability Statement:** All data included in this study are available upon request by contacting the corresponding author.

**Conflicts of Interest:** The authors declare no conflict of interest.

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
