# Peer review of "Coherent Multi-Dwell Processing of Un-Synchronized Dwells for High Velocity Estimation and Super-Resolution in Radar"

_remotesensing, doi:10.3390/rs15030782_

Round 1

Reviewer 1 Report

The manuscript “Coherent Multi-Dwell Processing of Unsynchronized Dwells for High Velocity Estimation and Super-Resolution in Radar” presented a multi-dwell processing method and compared it with classical radar processing methods, and the results showed improved performance.

The manuscript is overall well written, though a few clarifications and revisions are needed. I recommend publication with minor revision after the following comments are addressed:

1.     Line 220: The authors need to clarify which interpolation method was used. The “Common interpolation method” is not an interpolation method. Is it spline interpolation, linear interpolation, nearest neighbor method, IDW, Kriging, or something else?

2.     Line 223-230: The authors mentioned the error introduced by doing the interpolation, so like my previous comment, it’s important to mention what interpolation method was used and how the interpolation was done in the experiment. With different interpolation algorithms and different parameters, one can achieve very different results.

3.     Lines 77 -83 and lines 236-238: These paragraph line spaces are different from other paragraphs.

Author Response

The manuscript was revised in accordance with your remarks.

We hope that the revised paper meets all the requirements for publishing in Remote Sensing magazine.

Sincerely,

The authors.

Reviewer 2 Report

The authors manuscript „Coherent Multi-Dwell Processing of Unsynchronized Dwells 2 for High Velocity Estimation and Super-Resolution in Radar“ is interesting and necessary. This work certainly brings innovations and solutions to the most accurate detection of an object velocity. In addition, the work is dedicated to radars for UAVs, but it could probably used in the automotive industry for more accurate detection of pedestrians, cyclists and other moving objects.

After reading the work, it is definitely seen that it still needs to be worked on and improved.

It is not completely clear with the novelty of the work, because after reading it, it is not immediately obvious how the new coherent processing across multiple dwells in radar for single and multiple targets affects the improvement of the Doppler resolution? Besides, I ask authors to clearly indicate how velocity super resolution was achieved because in the 3 section was talked about accuracy. Either unify the terminology throughout the work or provide appropriate clarifications.

Some not critical text formatting mistakes throughout the paper appears, for example, values and units needs separation, such as fc = 3 GHz, or table 1 name is in page 5 but table is in page 6 and so on.

Furthermore, figure 4 and its explanation in one sentence and is not informative and needs more detailed explanation and updated figure for clarification.

It is hard to understand if interpolation drawback is significant or not from section 3.2. Maybe it is not significant but much more easily implemented? With explanation It is similar situation as previous note with the figure 4.

In the text ir it only noted what “The resolution improves when the target is illuminated for a longer time (line 169)" for a longer time” however never more is discussed about it and how this thesis is related to experiment simulations and provided results?

Author Response

(The authors gave the same response as above.)

Round 2

Reviewer 2 Report

Authors made additional improvements and I have no more comments